# Transcriptomics Reveals Effect of *Pulsatilla* Decoction Butanol Extract in Alleviating Vulvovaginal Candidiasis by Inhibiting Neutrophil Chemotaxis and Activation via TLR4 Signaling

**DOI:** 10.3390/ph17050594

**Published:** 2024-05-07

**Authors:** Hui Wu, Can Li, Yemei Wang, Mengxiang Zhang, Daqiang Wu, Jing Shao, Tianming Wang, Changzhong Wang

**Affiliations:** 1School of Integrated Traditional and Western Medicine, Anhui University of Chinese Medicine, Hefei 230012, China; wuhui2474170599@outlook.com (H.W.); eziooo2022@gmail.com (C.L.); wangyemei@ahtcm.edu.cn (Y.W.); tojosaki@126.com (M.Z.); daqwu@126.com (D.W.); ustcnjnusjtu@126.com (J.S.); wtm1818@163.com (T.W.); 2Institute of Integrated Traditional Chinese and Western Medicine, Anhui Academy of Chinese Medicine, Hefei 230012, China

**Keywords:** vulvovaginal candidiasis, *Candida albicans*, transcriptome assays, n-butanol alcohol extract of *Pulsatilla* decoction, TLRs/MyD88, neutrophil chemotaxis

## Abstract

The Pulsatilla decoction is a well-known herbal remedy used in clinical settings for treating vulvovaginal candidiasis (VVC). However, the specific mechanism that makes it effective is still unclear. Recent studies have shown that in cases of VVC, neutrophils recruited to the vagina, influenced by heparan sulfate (HS), do not successfully engulf *Candida albicans* (*C. albicans*). Instead, they release many inflammatory factors that cause damage to the vaginal mucosa. This study aims to understand the molecular mechanism by which the n-butanol extract of Pulsatilla decoction (BEPD) treats VVC through transcriptomics. High-performance liquid chromatography was used to identify the primary active components of BEPD. A VVC mouse model was induced using an estrogen-dependent method and the mice were treated daily with BEPD (20 mg/kg, 40 mg/kg, and 80 mg/kg) for seven days. The vaginal lavage fluid of the mice was analyzed for various experimental indices, including fungal morphology, fungal burden, degree of neutrophil infiltration, and cytokines. Various assessments were then performed on mouse vaginal tissues, including pathological assessment, immunohistochemistry, immunofluorescence, Western blot (WB), quantitative real-time PCR, and transcriptome assays. Our results showed that BEPD reduced vaginal redness and swelling, decreased white discharge, inhibited *C. albicans* hyphae formation, reduced neutrophil infiltration and fungal burden, and attenuated vaginal tissue damage compared with the VVC model group. The high-dose BEPD group even restored the damaged vaginal tissue to normal levels. The medium- and high-dose groups of BEPD also significantly reduced the levels of IL-1β, IL-6, TNF-α, and LDH. Additionally, transcriptomic results showed that BEPD regulated several chemokine (CXCL1, CXCL3, and CXCL5) and S100 alarmin (S100A8 and S100A9) genes, suggesting that BEPD may treat VVC by affecting chemokine- and alarmin-mediated neutrophil chemotaxis. Finally, we verified that BEPD protects the vaginal mucosa of VVC mice by inhibiting neutrophil recruitment and chemotaxis in an animal model of VVC via the TLR4/MyD88/NF-κB pathway. This study provides further evidence to elucidate the mechanism of BEPD treatment of VVC.

## 1. Introduction

*Candida albicans* (*C. albicans*) is a normal part of the female vaginal flora and does not normally cause infection. Long-term use of broad-spectrum antibiotics, glucocorticosteroids, or immunosuppressive drugs can cause changes in the vaginal environment that allow *C. albicans* to take advantage of the opportunity to reproduce in large quantities and invades the vaginal mucosa, leading to the occurrence of vulvovaginal candidiasis (VVC) [1,2]. Clinical symptoms include white soya bean dregs or curd-like vaginal discharge and redness, swelling, and itching of the vulva and vagina. This is often accompanied by pain during urination and sexual intercourse, seriously impacting the quality of life of patients with VVC [3,4]. Approximately 70–75% of women worldwide have experienced VVC at least once in their lifetime, with approximately 40–45% having experienced it two or more times [5].

Inherent immunity plays a more important role in fungal infections compared with adaptive immunity [6,7]. Neutrophils are important components of inherent immunity, and VVC is often closely associated with the concomitant recruitment of neutrophils to the vaginal lumen [8]. Notably, neutrophils are recruited by the production of heparan sulfate by vaginal epithelial cells. Instead of phagocytosing *C. albicans*, they release high levels of inflammatory factors, contributing to immune-inflammatory damage to the vaginal mucosa, termed “neutrophil anergy” [9]. Given that neutrophil recruitment to the vaginal lumen is driven by chemokines, blocking neutrophil exudation into the vaginal lumen by targeting chemokines is a novel strategy for treating VVC.

Currently, azoles, polyenes, and arylamines are most commonly used for the clinical treatment of VVC, with the advantages of short treatment periods and noticeable short-term efficacy. However, with the repeated use of these drugs, drug-resistant strains of fungi and other problems have become increasingly prominent, posing significant challenges to the treatment of VVC [10]. Traditional Chinese medicine (TCM) offers unique advantages in VVC treatment. It can directly inhibit the fungus and modulate the host’s immune response. Therefore, it is crucial to explore anti-VVC drugs from Chinese medicine resources and understand their immunomodulatory mechanisms. 

The *Pulsatilla* decoction is a TCM formulation composed of *Anemone chinensis*, *Phellodendron chinense* C.K. Schneid, *Coptis chinensis* Franch, and *Fraxinus chinensis* subsp. *rhynchophylla* (Hance) A. E. Murray. It is effective in clearing heat, removing toxins, and preventing dysentery. This herbal decoction is therefore often used in clinical practice for dysentery and ulcerative colitis. Recent studies have shown that the n-butanol extract of *Pulsatilla* decoction (BEPD) not only significantly affects *C. albicans* in vitro but also shows promising results against VVC [11,12]. Each herb in the *Pulsatilla* decoction plays a specific role in treating fungal infection. *Anemone chinensis* has a wide range of antibacterial effects, and anemonin along with anemoside as its main ingredients inhibit Staphylococcus aureus and Escherichia coli growth and exert anti-inflammatory effects through downregulation of the TLR4/NF-κB/MAPK pathway [13]. *Coptis chinensis* Franch has been demonstrated to possess significant broad-spectrum antimicrobial effects. Its active ingredients are alkaloids, including berberine and jatrorrhizine. Berberine has been demonstrated to downregulate the expression of the mycelium-specific gene ECE1 and inhibit the morphological transformation of *C. albicans* in vitro [14]. Additionally, it has been shown to reduce the activity of phospholipaseA2 (PLA2) in neutrophils, and inhibit the chemotaxis of neutrophils [15]. *Phellodendron chinense* C.K. Schneid exhibits antifungal, anti-inflammatory, and antioxidant properties that are comparable to *Coptis chinensis* Franch. Additionally, its principal active ingredients also include phellodendrine. *Fraxinus chinensis* subsp. *rhynchophylla* (Hance) A. E. Murray has been demonstrated to exert a pronounced inhibitory effect on the formation of biofilm and bud tube of *C. albicans*. Coumarins such as esculin and aesculetin are the principal active components of antifungal herbs, and also exhibit anti-inflammatory properties [16]. In summary, the *Pulsatilla* decoction has been demonstrated to possess antimicrobial properties, to inhibit neutrophil chemotaxis, and to relieve inflammation.

Clinically, Pulsatilla decoction has been demonstrated to be an effective treatment for VVC and to significantly reduce associated symptoms. This consequently prompted us to conduct further research into the specific molecular mechanisms underlying the therapeutic effects of BEPD on VVC. To this end, we employed transcriptome sequencing of the vagina to identify differentially regulated genes. The transcriptome results demonstrated that BEPD regulated several chemokines and S100 alarm protein genes and restored normal gene expression in VVC mice. This suggests that BEPD may affect neutrophil chemotaxis-related pathways, potentially contributing to its therapeutic efficacy. This is consistent with the findings of our previous in vivo study, which demonstrated the effect of BEPD on neutrophil chemotaxis in the vaginal mucosa of mice with VVC. S100A8 and S100A9 are endogenous agonists of TLR4, which is agonized to stimulate NF-κB through MyD88 [17]. This drives neutrophils to produce a variety of cytokines and chemokines. Consequently, the present study sought to elucidate the mechanism of BEPD treatment of VVC by examining the impact of BEPD on neutrophil chemotaxis and the activation of the TLR4 pathway, with a focus on the transcriptome.

## 2. Results

### 2.1. Main Components of BEPD

To identify the main active compounds in BEPD, we examined BEPD using HPLC (Figure 1). Esuclin, esculetin, epiberberine, berberine, phellodendrine, jatrorrhizine, and anemoside B4 were selected for identification based on their respective BEPD criteria. The retention times, peak areas, and contents are listed in Table 1. Among these constituents, berberine, esuclin, and anemoside B4 were the most abundant.

### 2.2. BEPD Improved the Symptoms of VVC

Clinically, the vaginas of patients with VVC usually produce a large amount of white discharge, accompanied with redness. To determine whether the VVC model induced similar symptoms in the mouse vagina, we captured vaginal photographs of each mouse group. Control mice showed normal activity throughout the experimental period. White discharge with redness was observed at the vaginal opening of mice from the VVC model group. Following drug treatment, local vaginal signs improved, and secretions decreased in the low- and medium-dose BEPD groups (20 mL/kg and 40 mL/kg). However, slight redness persisted at the vaginal opening but was absent in the high-dose BEPD (80 mL/kg) and fluconazole groups. Overall, the general and local conditions of the vagina were significantly improved and tended to be similar to those of the control mice (Figure 2).

The ability to switch from the yeast to the mycelial state during VVC is a key feature of *C. albicans* pathogenicity [18]. Therefore, we used Gram staining to observe changes in *C. albicans* morphology at different stages. During the experiment, *C. albicans* was not detected in the vaginal lavage of mice from the control group. Conversely, a significant presence of elongated hyphae was observed in the model group. Post-treatment with low-dose BEPD, medium-dose BEPD, high-dose BEPD, and fluconazole led to a reduction in *C. albicans* quantity and shortened hyphal length, particularly evident in the high-dose BEPD and fluconazole groups, where the distribution of *C. albicans* hyphae was nearly eradicated (Figure 3). Simultaneously, the effect of BEPD on the fungal content of the vagina in VVC model mice was assessed through serial dilution to determine the number of *C. albicans* colonies present in the vaginal lavage fluid. With smear counting, we found that the fungal load remained at the same level across all groups, except for the control group on days 1, 3, and 7. On day 14, the model group still exhibited a large fungal load, whereas after BEPD and fluconazole treatments, the number of fungi decreased significantly in a dose-dependent manner (Figure 4).

### 2.3. BEPD Alleviated Vaginal Histopathology

To further observe vaginal injury in mice, we performed hematoxylin and eosin (HE) and periodic acid–Schiff (PAS) staining of vaginal tissue sections. The results showed that the vaginal mucosa in the control group was superficially covered with a thick stratum corneum and structurally intact, while in the VVC model group, the vaginal mucosa completely disappeared. A large number of infiltrating inflammatory cells were also observed and the stratum corneum completely disappeared. In the VVC model group, the increase in stratum corneum was gradually restored, and inflammatory cell infiltration was reduced with an increase in the dose of BEPD. In the high-dose and fluconazole groups, the stratum corneum returned to normal levels, and the structural areas of the mucosal tissue remained intact (Figure 5). PAS staining showed similar results. In the VVC model, a large number of hyphae and spores adhered to the surface of the vaginal mucosa compared to the control group, and the numbers of hyphae and spores were slightly reduced in the low-dose BEPD group. However, in the medium-dose BEPD group, mycelia and spores were significantly reduced, and the stratum corneum was repaired. Remarkably, in the fluconazole and high-dose BEPD groups, no mycelia or spores were detected, nearly resembling the findings in the control group (Figure 6). When VVC occurs, *C. albicans* first colonizes and penetrates the vaginal mucosa, resulting in damage to the vaginal epithelium, and BEPD can ameliorate this process by reducing pathological damage and protecting the vaginal mucosa.

### 2.4. BEPD Reduced the Levels of Pro-Inflammatory Cytokines from the Vagina

Immunoinflammatory damage during the development of VVC leads to elevated levels of lactate dehydrogenase (LDH) and the cytokines IL-6, TNF-α, and IL-1β in vaginal epithelial cells [9]. This experiment showed that the above factors were significantly increased in VVC model mice compared to those in control mice. LDH, which reflects cellular damage factors, and the inflammatory factors IL-6, TNF-α, and IL-1β were significantly reduced after BEPD treatment (Figure 7). These results suggest that BEPD reduces pro-inflammatory cytokines in the vagina, mitigating inflammation in vaginal tissues.

### 2.5. Effect of BEPD on the Transcriptomics Profile of VVC Mice

To further understand the potential molecular mechanisms underlying the regulation of vaginal barrier function and inflammation in VVC mice, we used the Illumina high-throughput sequencing platform to analyze the transcriptome sequences of animal tissues from the control, modeling, and high-dose BEPD groups of animal models. According to the principal component analysis (PCA, Figure 8A), the differences among the three groups were significant, suggesting that modeling and pharmacological interventions can affect transcriptional patterns. Compared to the control group, genes in the model group were significantly shifted to the left; however, the BEPD group tended to return to normal, suggesting that BEPD can suppress gene expression under VVC conditions.

We focused on the genes that were downregulated in the model group compared to those in the BEPD group. Comparing the model group to the control group, there were 3014 differentially expressed genes (1199 upregulated and 1815 downregulated), whereas there were 3045 differentially expressed genes (1411 upregulated and 1634 downregulated) in the model group compared to the BEPD group (Figure 8B,C). This suggests that more proteins were expressed by the upregulation of these genes in the VVC model, and expression was reduced after BEPD treatment. Importantly, according to Kyoto Encyclopedia of Genes and Genomes (KEGG) functional annotation analysis, the differentially expressed genes were mainly enriched in signaling molecules and interaction, signal transduction, and the immune system (Figure 8D). Gene Ontology (GO) enrichment analysis showed the top 20 enrichment pathways in which these differentially expressed genes were collectively involved (Figure 8E). Among these top 20 enriched pathways, it was found that BEPD restored normal gene expression in VVC mice, which could be through affecting neutrophil chemotaxis-related pathways. Further clustering analysis of the related genes, as shown in the heatmap (Figure 8F), revealed that BEPD regulated several chemokines as well as S100 alarm protein genes. These differentially expressed genes were further verified by RT-qPCR. As shown in Figure 8G, the mRNA expression of relevant chemokines and S100 genes was significantly increased in the model group compared to the control group, and the expression of these differentially expressed genes was significantly reduced after BEPD treatment.

### 2.6. Effect of BEPD on Neutrophil Activity in the Vaginal Mucosal Tissue of VVC Mice

Studies have shown that when VVC occurs, *C. albicans* can activate innate immune signals, leading to neutrophil recruitment to the vaginal mucosa. The neutrophil influx exacerbates the inflammatory cascade, resulting in chronic inflammation of the vaginal mucosa [19,20]. The results of Papanicolaou (Pap) staining showed that a substantial number of neutrophils were present in the vaginal lavage fluid from VVC model mice compared to that from control mice. Additionally, BEPD (40 and 80 mL/kg) and fluconazole treatment significantly reduced neutrophil levels (Figure 9). This suggests that BEPD can reduce the accumulation and formation of neutrophils and inhibit their activity. Immunofluorescence (IF) staining for the common neutrophil marker lymphocyte antigen 6 complex G6D (Ly6G) and immunohistochemistry (IHC) for myeloperoxidase (MPO) demonstrated a significant increase in neutrophil recruitment in the vaginal tissues of VVC mice compared to controls. However, BEPD administration resulted in decreased neutrophil accumulation in the tissues (Figure 10 and Figure 11) [21,22]. The fluconazole group showed almost no difference compared with the control group. The results indicate that BEPD inhibits neutrophil activity in the vaginal mucosal tissue of VVC mice.

### 2.7. Effect of BEPD on Chemokines in the Vaginal Mucosal Tissue of VVC Mice

Neutrophil elastase (NE) is a serine protease stored in neutrophil progenitor granules that promotes tissue infiltration of neutrophils. NE acts as a pro-inflammatory mediator by regulating cytokine and chemokine activity through the process of protein hydrolysis. NE was labeled for IHC to observe changes within vaginal tissues (Figure 12). In the model group, protein expression of NE was significantly elevated compared to the control group. Notably, the low-dose BEPD group exhibited levels akin to the model group, while the fluconazole and high-dose BEPD groups gradually approached levels similar to the control group. This not only indicates that BEPD reduced the infiltration of neutrophils but also further suggests that the activity of chemokines was inhibited. Therefore, we analyzed the protein expression of the mouse vaginal tissue chemokines CXCL1, CXCL3, and CXCL5 using Western blotting. Among the three doses of BEPD, the highest dose (80 mL/kg) was the most effective in reducing the expression of these proteins (Figure 13). Additionally, S100 Alin (alarmin), produced by phagocytes, monocytes, epithelial cells, and endothelial cells [23,24,25], is released at inflammation sites. Recent evidence suggests that neutrophile migration is mediated by chemotactic S100A8 and S100A9 signals generated during the response of vaginal epithelial cells to *Candida* [17]. Western blot results also indicated that S100A8 and S100A9 protein expression gradually decreased with an increase in BEPD dose compared to the model group (Figure 13). In summary, these results indicate that BEPD reduces chemokine expression and inhibits neutrophil chemotaxis in the vaginal mucosal tissues of VVC mice.

### 2.8. BEPD Downregulated the TLR4/MyD88/NF-κB Signaling Pathway in VVC Mice

TLR signaling is believed to be involved in neutrophil recruitment coordinated by chemokines and inflammatory cytokines, which subsequently leads to the direct killing of invading pathogens [26]. TLR4 receptors are specifically present in multiple cell types and play important roles in regulating inflammation [27]. In VVC, TLR4 aggregation is induced to recruit MyD88, which activates the downstream NF-κB signaling pathway to induce inflammatory cytokine release. To assess the effect of BEPD on the TLR4 pathway, we determined the expression of proteins involved in the TLR4 pathway and found that the protein expression of TLR4, MyD88, and NF-κB was significantly reduced with increasing BEPD doses. These results indicate that BEPD inhibited the activation of the TLR4 pathway (Figure 14). We then examined the co-expression of Ly6G with TLR4 and MyD88 using immunofluorescent colocalization. In VVC mice, Ly6G exhibited high co-expression with TLR4 and MyD88, which decreased following BEPD treatment (Figure 15 and Figure 16). These results indicate that neutrophil chemotaxis and its activation were inhibited, which may be related to the fact that BEPD can downregulate TLR4/MyD88/NF-κB signaling in VVC.

## 3. Discussion

VVC is an inflammatory injury caused by *C. albicans* to the mucous membrane of the female genital tract. Although VVC infection is not life-threatening, its recurrent nature is physically and mentally debilitating [28]. Clinical treatment of VVC mainly relies on azole antifungal drugs, such as fluconazole, which achieve therapeutic effects by targeting the synthesis pathway of ergosterol on fungal cell membranes to inhibit the organism. However, the long-term application of such drugs leads to the development of drug resistance in *C. albicans*. Moreover, resistance to other antifungal drugs and potential toxic side effects on organs, such as the liver and kidney, are concerns that cannot be overlooked.

In this study, we investigated the efficacy and underlying mechanisms of BEPD in treating VVC mice. We found that BEPD regulated neutrophil recruitment and chemotaxis by downregulating the TLR4 signaling pathway, thereby protecting the vaginal mucosa of VVC mice. Redness and swelling of the vaginal opening with a white mucoid secretion covering is a typical feature of VVC mice, and clinically, the same symptoms occur in the vaginas of patients with VVC. In this study, we demonstrated that BEPD ameliorated these symptoms in the mouse vagina. The vaginal epithelial barrier is a tightly connected epithelial cell that forms a line of defense against external pathogens invading the vagina [29,30,31]. During VVC, the vaginal mucosal barrier function is impaired. Simultaneously, *C. albicans* undergoes morphological transformation from yeast to mycelium [32]. The mycelium form invades the submucosa through the mucosa or through gaps between epithelial cells. Our results suggest that BEPD effectively attenuates the burden of *C. albicans*. Gram and PAS staining revealed that mycelial formation and *C. albicans* adhesion were reduced in the BEPD-treated group. HE staining revealed that compared to the normal group of mice, the model group exhibited squamous epithelial cell proliferation, disruption of the mucosal layer, and infiltration of inflammatory cells. However, in the medium- and high-dose BEPD treatment groups, the mucosal layer was restored, and inflammatory infiltration was significantly reduced. These results indirectly indicate that BEPD could attenuate the inflammatory response in VVC mice and effectively inhibit vaginal barrier damage.

VVC is primarily a mucosal infectious disease where innate immunity plays a crucial role in defense. Neutrophils, key components of innate immunity, are vital in controlling fungal infections and are among the first cells recruited to sites of inflammation [33]. However, in VVC, neutrophils not only fail to clear the pathogen *C. albicans* but also become activated upon recruitment to the vaginal mucosa [34]. Activated neutrophils release pro-inflammatory cytokines, such as TNF-α, IL-1β, and IL-6, exacerbating inflammation instead of providing protection as observed in infection and tissue injury scenarios. In the present study, we demonstrated that the levels of pro-inflammatory cytokines in the neutrophile and vaginal lavage fluid of mice were significantly reduced after BEPD treatment, further verifying the anti-inflammatory function of BEPD. IF and IHC showed that Ly6G and MPO expression was significantly reduced in the BEPD-treated group compared to that in the model group. This indicates that BEPD reduced the number of granulocytes and significantly limited neutrophil activation and infiltration into the vaginal tissues. In addition, neutrophil recruitment is closely associated with various chemokines. Among them, chemokines, such as CXCL1, CXCL3, and CXCL5, play key roles in promoting neutrophil migration and chemotaxis by interacting with CXCR2 receptors [35]. Interestingly, CXCL1, CXCL3, and CXCL5 proteins were highly expressed in the vaginal tissues of VVC mice, whereas the drug-treated group presented opposite results. These findings suggest that BEPD exerts protective effects in VVC mice by inhibiting neutrophil chemotaxis and reducing their recruitment to the vagina, thereby improving the clinical remediation of VVC.

To gain further insights into the underlying molecular mechanisms, we performed transcriptome analysis and screened for key genes regulated by BEPD in VVC mice. Interestingly, besides the chemokines known to promote neutrophil migration and aggregation, we discovered significant alterations in the expression of two major factors secreted by neutrophils, namely S100A8 and S100A9, in BEPD-treated VVC mice. When *C. albicans* infection occurs, S100 is produced at the site of vaginal epithelial cell inflammation, with *C. albicans* adherence to vaginal epithelial cells being critical for the induction of S100 proteins and the subsequent neutrophil response [21]. These findings suggest that neutrophil migration is mediated by the production of chemotactic S100A8 and S100A9 alert proteins by vaginal epithelial cells in response to *C. albicans* [36,37,38,39]. Researchers have further elucidated the biological role of S100 alert proteins in the pathogenesis of *C. albicans* vaginal infections, observing that antibody neutralization of S100A8 and S100A9 affects the chemotactic activity of neutrophil in vaginal lavage [40]. In our study, BEPD significantly downregulated the levels of S100A8, S100A9, and other proteins in the vaginal tissues of VVC mice. S100A8/A9 is an endogenous agonist of TLR4, and agonization of TLR4 stimulates the NF-κB-mediated production of IL-1β precursors via MyD88 [17]. In the present study, we found that BEPD reversed *C. albicans*-induced increases in TLR4, MyD88, and NF-κB expression. The results suggest that S100A8/A9 plays a molecular role in vaginal barrier injury in VVC mice. Additionally, the study elucidated that BEPD regulates vaginitis and enhances vaginal barrier function in VVC mice by inhibiting the release of S100A8/A9 and downregulating the TLR4/MyD88/NF-κB signaling pathway.

In conclusion, this study demonstrated that BEPD regulates neutrophil recruitment and chemotaxis by inhibiting the release of S100A8/A9 and downregulating the TLR4/MyD88/NF-κB signaling pathway, thereby protecting the vaginal mucosa of VVC mice (Figure 17). However, this study has some limitations. Although we identified several active components in BEPD that are believed to mediate its anti-inflammatory effects, further studies are needed to determine the specific targets and mechanisms of action of these components. Completion of such studies will provide valuable insights into the therapeutic utility of BEPD for VVC treatment.

## 4. Materials and Methods

### 4.1. Reagents

Fluconazole was purchased from Yuanye (JO9M6B1; Shanghai, China). Estradiol benzoate was obtained from Dibo Biotechnology Co., Ltd. (FM19; Shanghai, China). Solid and liquid YPD media were purchased from Qingdao High-Tech Industrial Park Haibo Biotechnology Co., Ltd. (HB5193 and HB5193-1, respectively; Qingdao, China). Gram staining fluid was purchased from Zhuhai Besso Biotech Co., Ltd. (C200901; Beijing, China). TNF-α, IL-6, IL-1β, and LDH Enzyme-linked immunosorbent assay (ELISA) kits were bought from Shanghai Jianglai Biotechnology Co., Ltd. (Shanghai, China). TLR4, MyD88, CXCL1, CXCL3, CXCL5, S100A8, and neutrophil elastase were purchased from Affinity (#AF7017, #AF5195, #AF5403, #DF8554, #DF9919, #DF6556, and #AF0010; Shanghai, China). NF-κB p65 and Ly6G were bought from Cell Signaling Technology (#6956 and #87048; Danvers, MA, USA). S100A9 was purchased from ZEN BIO (#R25648; Chengdu, China) and MPO were obtained from Proteintech (#R25648; Melbourne, Australia).

### 4.2. Strains and Drugs

*C. albicans* SC5314 was generously provided by Prof. Jiang Yuanying of the Second Military Medical University, China. Single colonies of *C. albicans* were picked from 4 °C preserved Sabouraud’s Dextrose Agar, inoculated into liquid YPD medium at 37 °C, and grown for 14–16 h. The supernatant was discarded following centrifugation at 3000× *g* for 5 min and resuspended in RPMI-1640 medium at pH 7.0.

The *Pulsatilla* decoction was composed of *Anemone chinensis* (15 g), *Phellodendron chinense* C.K.Schneid. (12 g), *Coptis chinensis* Franch. (6 g), and *Fraxinus chinensis* subsp. *rhynchophylla* (Hance) A. E. Murray (12 g). It was purchased from the Traditional Chinese Medicine Pharmacy of the First Affiliated Hospital of Anhui University of Traditional Chinese Medicine (Hefei, China) and was identified by Prof. Liu Shoujin.

### 4.3. Preparation of n-Butanol Alcohol Extract from Pulsatilla Decoction

The *Pulsatilla* decoction was subjected to extraction by soaking overnight in 80% ethanol, followed by heating in a water bath and refluxing at 70 °C for 3 h, and repeated three times. The collected filtrate was pooled for further analysis. The solution was extracted 5–6 times according to different polar gradient solutions (petroleum ether, ethyl acetate, n-butanol), and the combined solution was dried and evaporated at 80 °C. The final BEPD yield was 13.0–15.0%.

### 4.4. Qualitative Analysis of BEPD Extract via HPLC

Anemoside B4, phellodendrine, esculin, esculetin, epiberberine, berberine, and jatrorrhizine contents in BEPD were detected via HPLC. The BEPD solution was prepared by precisely weighing 0.2 g of the product powder, obtained through a second sieve, and placing it in a securely stoppered conical flask. A mixture of methanol and hydrochloric acid in a 100:1 ratio was then added to the flask, followed by careful weighing to determine the combined weight of the powder and solvent mixture. Ultrasound was applied for 30 min at a power of 250 W and a frequency of 40 kHz. After cooling, the flask was reweighed to account for any weight changes. If there was any observed loss, it was compensated for by adding methanol. This meticulous procedure ensured the accurate preparation of the BEPD solution, maintaining consistency and reliability in subsequent analyses. The chromatographic conditions and system suitability test utilized octadecylsilane-bonded silica gel as the stationary phase and a mobile phase composed of acetonitrile and 0.05 mol/L potassium dihydrogen phosphate solution in a 50:50 ratio. The mobile phase was further modified by adding 0.4 g of sodium dodecyl sulfate to every 100 mL, and the pH was adjusted to 4.0 using phosphoric acid.

### 4.5. Experimental Animals

C57BL/6 female mice (20–22 g; 6–8 weeks) were purchased from the Experimental Center of Anhui Medical University (Hefei, China). Mice were maintained in a disease-free environment, with a humidity of 50–55%, temperature of 22 ± 2 °C, and a 12 h light/dark cycle, and given free access to food and water. All animal procedures related to this experiment were approved by the Animal Ethics Committee of the Institute of Anhui University of Chinese Medicine and followed the guidelines of the Chinese legislation on the ethical use and care of laboratory animals.

### 4.6. Establishment and Treatment of the VVC Model

Seventy-two female C57BL/6 mice were randomly divided into six groups: control, VVC model, fluconazole, low- (20 mL/kg), medium- (40 mL/kg), and high-dose (80 mL/kg) BEPD. Except for mice in the control group, 0.1 mL of sesame oil containing 0.1 mg of estradiol benzoate was injected into the neck every other day [39,40]. Additionally, 20 μL of a *C. albicans* suspension at a concentration of 2.5 × 10^8^ cells/mL was inoculated in the mouse vagina for 1 week before further treatment. The VVC model group received saline gavage, while the fluconazole group received 20 mg/kg fluconazole and 20, 40, and 80 mg/kg BAEB once a day for 7 d. Vaginal lavage stains were collected on days 1, 3, 7, and 14 after infection, and the VVC model was observed and identified under light microscopy.

### 4.7. Gram Staining

Approximately 10 μL of lavage fluid was transferred to slides, spread evenly using a pipette tip, and stained using a Gram stain kit (Beijing Solarbio Science & Technology Co., Ltd., Beijing, China) [12]. Fungal morphology was observed and recorded using a microscope (Olympus BX51; Fulai Optical Technology Co. Ltd., Shanghai, China) at 200× magnification.

### 4.8. Vaginal Local Characteristics and CFU Count

Fungal loading in mouse vaginal lavage fluid was determined using the plate method. Specifically, 30 μL of mouse vaginal lavage fluid was diluted in PBS, coated on YPD agar medium, and incubated in a 37 °C incubator for 24 h to quantify the fungal load (CFU/mL) [39].

### 4.9. Neutrophil Count of the Vaginal Lavage Fluid

At the end of the treatment period, 30 μL of lavage solution was evenly transferred onto slides and fixed for 15 min in 95% ethanol before staining using a Pap staining kit [12]. Neutrophils were identified under a light microscope (Olympus BX51; Fulai Optical Technology Co. Ltd., Shanghai, China) at 200× magnification.

### 4.10. Histopathology Analysis

Dissected vaginal tissues were fixed in 10% neutral-buffered formalin for more than 24 h before being dehydrated using a gradient of ethanol and embedded in paraffin. The tissue was cut into 4 µm thick sections using a microtome and stained using PAS and HE kits [41].

### 4.11. ELISA

Mouse vaginal lavage fluid was pipetted into sterile EP tubes and centrifuged at 3000× *g* for 20 min. Subsequently, ELISA kits (Jianglai Biotechnology Co. Ltd., Shanghai, China) were used to detect IL-1 β, LDH, IL-6, and TNF-α in the supernatant [12].

### 4.12. IHC

Well-sliced vaginal tissues were dewaxed with xylene, rehydrated with gradient ethanol, and treated with sodium citrate buffer (10 mM, PH 6.0) for 20 min [42]. After cool-down to room temperature, the tissue was incubated in 3% hydrogen peroxide for 10 min to inactivate endogenous peroxidase. Tissues were then incubated with 5% BSA for 20 min and incubated with primary antibodies against MPO and NE overnight at 4 °C. Tissues were washed overnight with PBS at room temperature, followed by titration with biotinylated goat anti-mouse IgG and SABC. Development with DAB peroxidase, restaining with hematoxylin, sealing with neutral resin, and examination under light microscopy (Olympus BX51; Fulai Optical Technology Co. Ltd., Shanghai, China) at 200× magnification were conducted. Image analysis was performed using Image J.

### 4.13. IF

The protein expression levels of Ly6G, TLR4, MyD88, and NF-κB were determined via IF staining. Briefly, sliced vaginal tissues were dewaxed, rehydrated, and treated with sodium citrate buffer for 20 min [43]. After cooling to room temperature, tissue was infiltrated with 0.5% Triton X-100 (Cat#T8200; Beijing Solarbio Science & Technology Co., Ltd., Beijing, China) in PBS, blocked with goat serum, and incubated overnight with 4 °C with antibodies against TLR4, MyD88, and NF-κB. The membranes were incubated with fluorescent secondary antibodies for 1 h, washed, and subsequently incubated overnight with antibodies against Ly6G. After overnight incubation, the tissues were removed and incubated with another fluorescently labeled secondary antibody. The final sections were counterstained with 4′, 6-diamine-2′-benzendol (DAPI) for 5 min. IF was observed using a laser confocal microscope (DMi8; Leica Microsystems, Wetzlar, Germany).

### 4.14. Transcriptome Sequencing

Total RNA for transcriptome sequencing was extracted from the vaginal tissue of control (*n* = 3), model (*n* = 3), and BEPD-H (*n* = 3) mice. The concentration, purity, and integrity of the extracted RNA were examined using a Nanodrop2000 system (www.majorbio.com) and agarose gel electrophoresis. After A-T base pairing with poly-A using magnetic beads with oligo (d-T), mRNA was isolated from total RNA. The resulting mRNA was added to the fragmentation buffer to isolate a small fragment of approximately 300 bp for the subsequent reverse synthesis of cDNA. After forming the cDNA into a stable double-stranded structure, the adaptor was ligated. Sequencing was performed using the Illumina NovaSeq 6000 (www.majorbio.com, 1 January 2024) sequencing platform. Genes with a threshold of |log2FC|> 1 and a *p*-value <0.05 were selected. Gene sets from Gene Ontology (GO) and Kyoto Encyclopedia of Genes and Genomes (KEGG) were further analyzed using gene set enrichment analysis to interpret the gene expression data. The purpose of this step was to observe the trends in the relevant pathways and functions involved in differential gene expression.

### 4.15. Western Blot

Vaginal tissue weighing 0.1 g was homogenized in RIPA lysis buffer. The homogenate was then centrifuged at 4 °C, 3000 rpm for 5 min, and the supernatant was collected. Total protein was obtained by boiling the protein buffer. Proteins were separated via SDS-PAGE and transferred onto PVDF membranes using a transmembrane device (WIX-miniPRO, WIX Technology Co., Ltd., Beijing, China). They were then blocked with a 5% skim milk solution. Subsequently, the bands were incubated with the corresponding primary antibodies for CXCL1, CXCL3, CXCL5, S100A8, S100A9, MyD88, NF-κB, and TLR4 at a ratio of 1:1000 overnight. The following day, the blots were thoroughly washed with PBS-Tween 20 and incubated with secondary antibodies for 1 h at room temperature. Antigen-antibody binding was detected using an enhanced chemiluminescence substrate kit. The bands were analyzed using ImageJ software.

### 4.16. Quantitative Real-Time PCR Analysis

One hundred milligrams of frozen vaginal tissue was combined with one milliliter of SparkZol and ground in liquid nitrogen. The homogenized samples were vigorously shaken and then left at room temperature for five minutes. The top aqueous phase was extracted by centrifugation with two hundred microliters of chloroform. To obtain total RNA, the samples were combined with an equal volume of isopropanol, centrifuged for 10 min, and the supernatant was discarded. Then, 75% ethanol was added to wash the precipitate and centrifuged to discard the supernatant. Finally, an appropriate quantity of RNase-free water was added to the air-dried samples in order to dissolve them and obtain RNA. The total RNA was extracted and reverse-transcribed into cDNA in accordance with the instructions provided in the reverse transcription kit (SparkJade, Jinan, China). PCR was performed using the Fluorescent Quantitation PCR System ABI 7500 (Applied Biosystems, Waltham, MA, USA). The experimental conditions were as follows: pre-denaturation at 95 °C for 60 s, followed by 40 cycles at 94 °C for 18 s, 57 °C for 18 s, and 72 °C for 25 s. The primer sequences are shown in Table 2. The relative expression of target genes was calculated using the 2^−ΔΔCT^ method. 

### 4.17. Statistical Analysis

All data were analyzed using SPSS 23.0 statistical software. Measurement data were expressed as mean ± SD, and differences between groups were compared using a one-way ANOVA test, with *p* < 0.05. The experiments were repeated thrice.

## 5. Conclusions

In this study, we systematically confirmed the therapeutic effects of BEPD on protein expression and pathological conditions in VVC mice. The results showed that BEPD acted in a dose-dependent manner in VVC mice. Specifically, BEPD mitigated VVC symptoms and safeguarded vaginal tissues by inhibiting neutrophil chemotaxis and activation, as well as downregulating the TLR4/MyD88/NF-κB pathway. This study sheds light on the pharmacological mechanism underlying BEPD’s action in VVC and strengthens the evidence supporting its utilization in VVC treatment.

## Figures and Tables

**Figure 1 pharmaceuticals-17-00594-f001:**
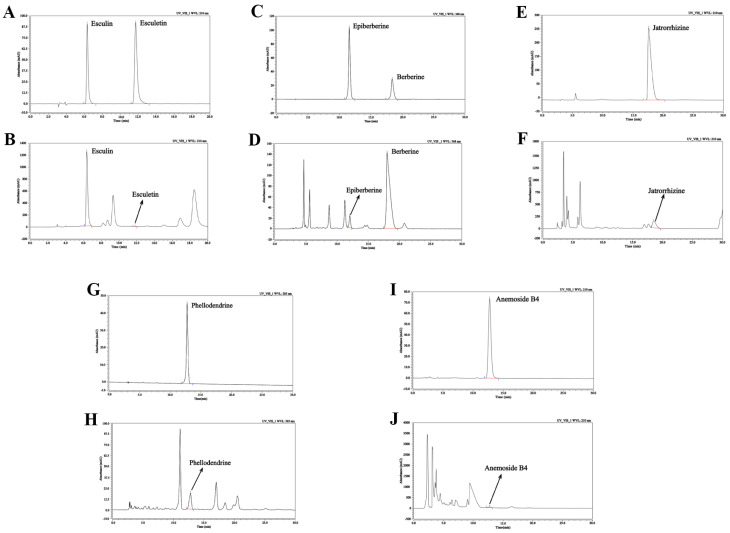
HPLC fingerprint of major components of BEPD. (**A**,**C**,**E**,**G**,**I**) HPLC fingerprint of the esculin, esculetin, epiberberine, berberine, jatrorrhizine, phellodendrine, and anemoside B4 standards. (**B**,**D**,**F**,**H**,**J**) HPLC fingerprint of the esculin, esculetin, epiberberine, berberine, jatrorrhizine, phellodendrine, and anemoside B4 of BEPD. HPLC, high-performance liquid chromatography; BEPD, the n-butanol extract of Pulsatilla decoction.

**Figure 2 pharmaceuticals-17-00594-f002:**
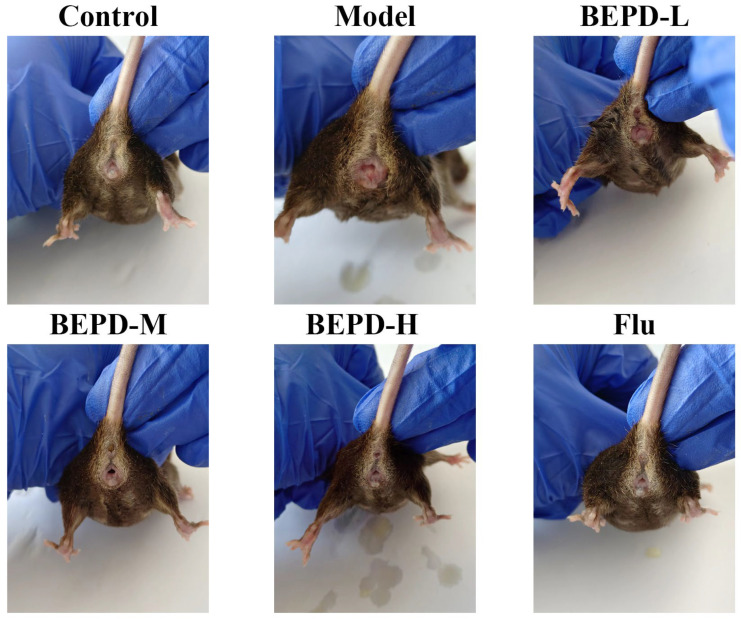
Reducing vaginal inflammation in mice by BEPD intervention for *C. albicans*-induced inflammatory vulvovaginal candidiasis symptoms. Images acquired on the seventh day of BEPD treatment. BEPD, the n-butanol extract of Pulsatilla decoction; BEPD-L, low-dose BEPD group (20 mL/kg); BEPD-M, medium-dose BEPD group (40 mL/kg); BEPD-H, high-dose BEPD group (80 mL/kg); Flu, fluconazole group.

**Figure 3 pharmaceuticals-17-00594-f003:**
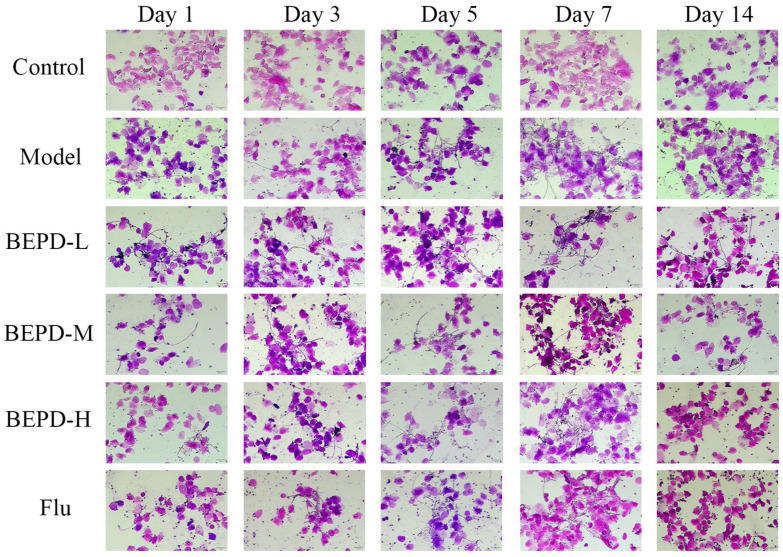
BEPD inhibited vaginal colonization of *C. albicans* in mice. Images represent Gram staining of vaginal lavage fluid from mice infected with *C. albicans* on days 1, 3, and 7 and treated with BEPD on day 14. BEPD, n-butanol extract of Pulsatilla decoction; BEPD-L, low-dose BEPD group (20 mL/kg); BEPD-M, medium-dose BEPD group (40 mL/kg); BEPD-H, high-dose BEPD group (80 mL/kg); Flu, fluconazole group.

**Figure 4 pharmaceuticals-17-00594-f004:**
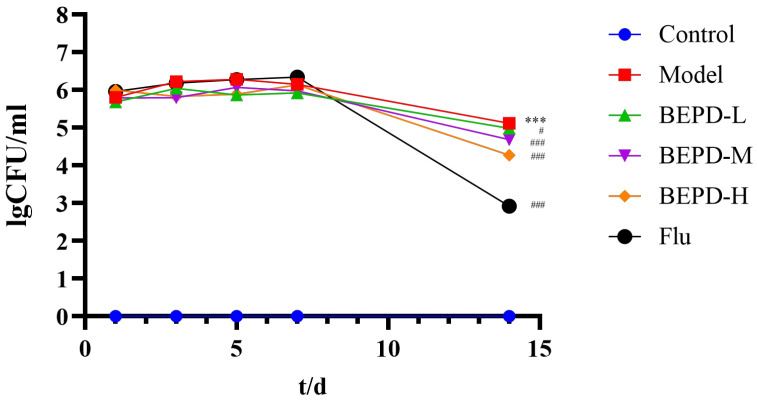
BEPD reduced the fungal load in the vaginas of mice with VVC. Figure 4 shows the fungal loads in the vaginas of mice on days 1, 3, 5, and 7 after *C*. *albicans* stimulation and on day 7 after intervention. Values are presented as mean ± SD. *n* = 3. *** *p* < 0.001, vs. control group; # *p* < 0.05, vs. model group; ### *p* <0.001, vs. model group. BEPD, n-butanol extract of Pulsatilla decoction; BEPD-L, low-dose BEPD group (20 mL/kg); BEPD-M, medium-dose BEPD group (40 mL/kg); BEPD-H, high-dose BEPD group (80 mL/kg); Flu, fluconazole group; VVC, vulvovaginal candidiasis.

**Figure 5 pharmaceuticals-17-00594-f005:**
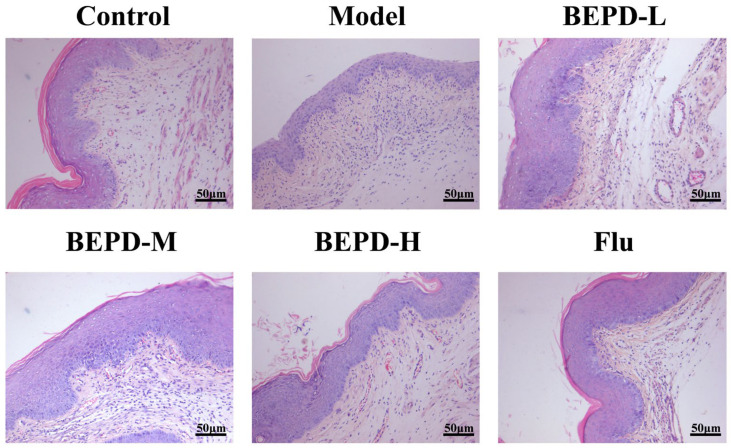
BEPD reduces vaginal symptoms, promotes stratum corneum repair, improves vaginal tissue damage, and reduces inflammatory cell infiltration in mice with VVC (H&E staining, 200×). BEPD, n-butanol extract of Pulsatilla decoction; BEPD-L, low-dose BEPD group (20 mL/kg); BEPD-M, medium-dose BEPD group (40 mL/kg); BEPD-H, high-dose BEPD group (80 mL/kg); Flu, fluconazole group; VVC, vulvovaginal candidiasis; H&E, hematoxylin and eosin.

**Figure 6 pharmaceuticals-17-00594-f006:**
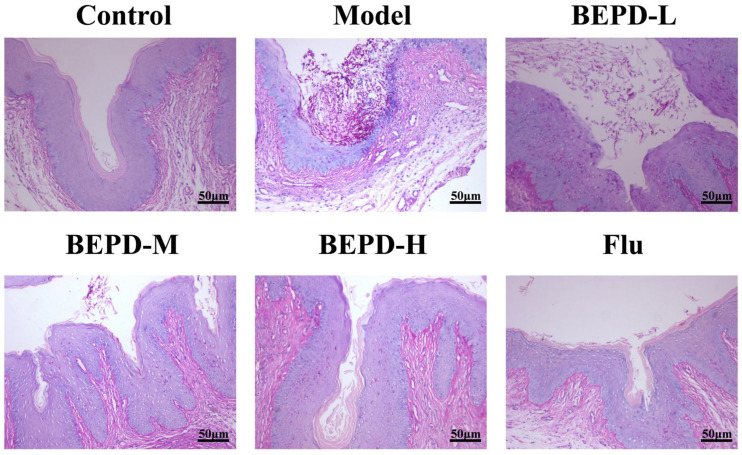
Effect of BEPD on vaginal histopathology of VVC mice induced by C. albicans. BEPD inhibits the adhesion and colonization of C. albicans on the vaginal mucosa in VVC mice. (PAS staining; 200×.) BEPD, n-butanol extract of Pulsatilla decoction; BEPD-L, low-dose BEPD group (20 mL/kg); BEPD-M, medium-dose BEPD group (40 mL/kg); BEPD-H, high-dose BEPD group (80 mL/kg); Flu, fluconazole group; VVC, vulvovaginal candidiasis; PAS, periodic acid–Schiff.

**Figure 7 pharmaceuticals-17-00594-f007:**
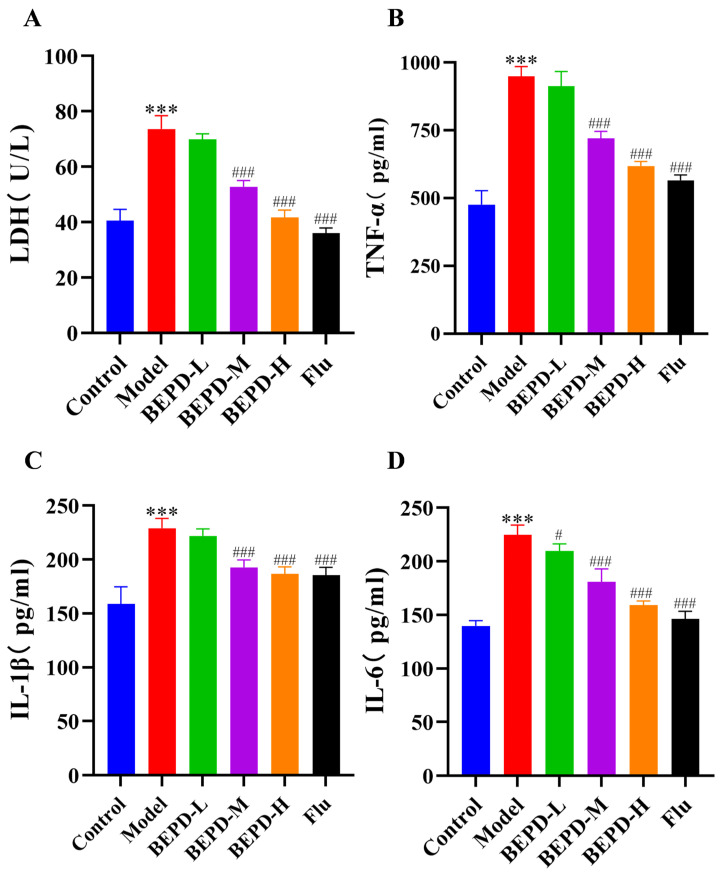
BEPD decreased the release of IL-1β, TNF-α, IL-6, and LDH from vaginal lavage fluid in VVC mice. (**A**) LDH, (**B**) TNF-α, (**C**) IL-1β, and (**D**) IL-6 cytokine levels were detected using ELISA. Data are the mean ± SD (*n* = 3). *** *p* < 0.001 versus the control group; # *p* < 0.05, ### *p* < 0.001 versus the model group. Values are presented as mean ± SD. n = 3. *** *p* < 0.001, vs. control group; # *p* < 0.05, vs. model group; ### *p* < 0.001, vs. model group. BEPD, n-butanol extract of Pulsatilla decoction; BEPD-L, low-dose BEPD group (20 mL/kg); BEPD-M, medium-dose BEPD group (40 mL/kg); BEPD-H, high-dose BEPD group (80 mL/kg); Flu, fluconazole group; VVC, vulvovaginal candidiasis; IL, interleukin.

**Figure 8 pharmaceuticals-17-00594-f008:**
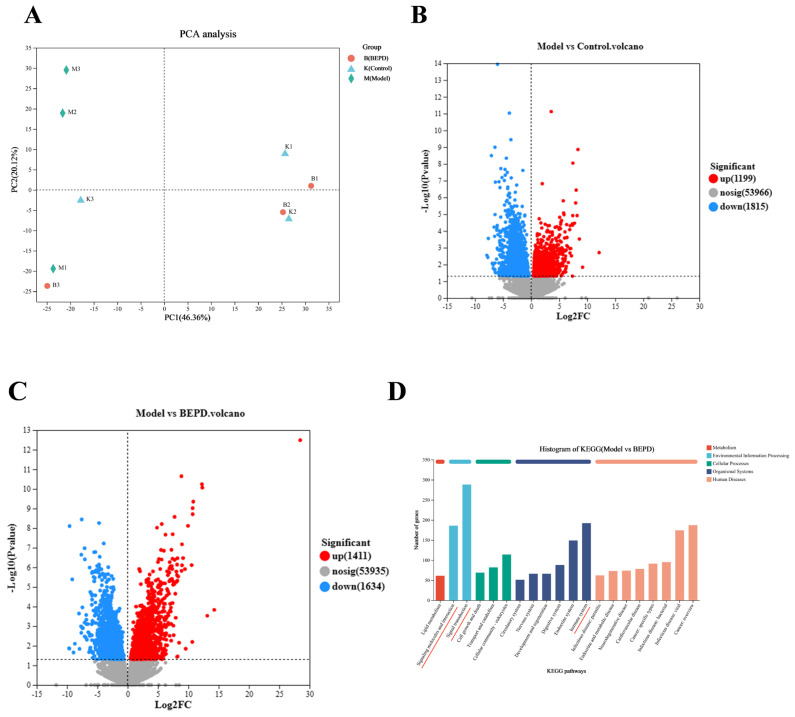
BEPD regulates the expression of genes associated with neutrophil chemotaxis. (**A**) PCA mapping of control, model, and BEPD samples. (**B**) Volcano plots between model and control groups. (**C**) Volcano plots between model and BEPD groups. (**D**) Annotation analysis of KEGG function in the model group vs. BEPD group. (**E**) GO functional enrichment analysis of the model group vs. BEPD group. (**F**) Heatmap of clustering analysis of neutrophil chemotaxis-related genes. (**G**) The mRNA expression of CXCL1, CXCL3, CXCL5, S100A8, and S100A9 in vaginal tissues. Values are presented as mean ± SD. n = 3. *** *p* < 0.001, vs. control group; ### *p* < 0.001, vs. model group. BEPD, n-butanol extract of Pulsatilla decoction; GO, Gene Ontology; KEGG, Kyoto Encyclopedia of Genes and Genomes.

**Figure 9 pharmaceuticals-17-00594-f009:**
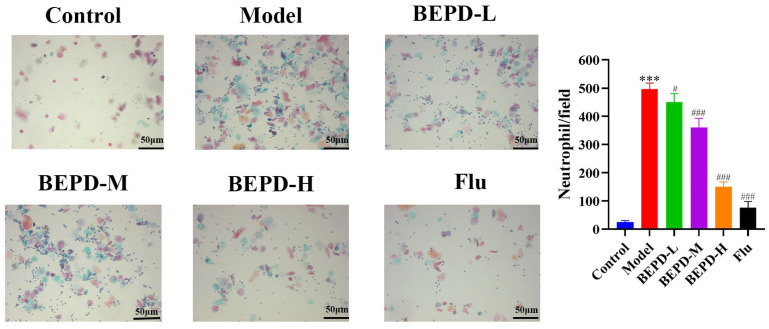
BEPD reduces neutrophil accumulation in mice with VVC. (Pap staining, 200×.) Values are presented as mean ± SD. n = 5. *** *p* < 0.001, vs. control group; # *p* < 0.05, vs. model group; ### *p* < 0.001, vs. model group. BEPD, n-butanol extract of Pulsatilla decoction; BEPD-L, low-dose BEPD group (20 mL/kg); BEPD-M, medium-dose BEPD group (40 mL/kg); BEPD-H, high-dose BEPD group (80 mL/kg); Flu, fluconazole group; VVC, vulvovaginal candidiasis; Pap, Papanicolaou.

**Figure 10 pharmaceuticals-17-00594-f010:**
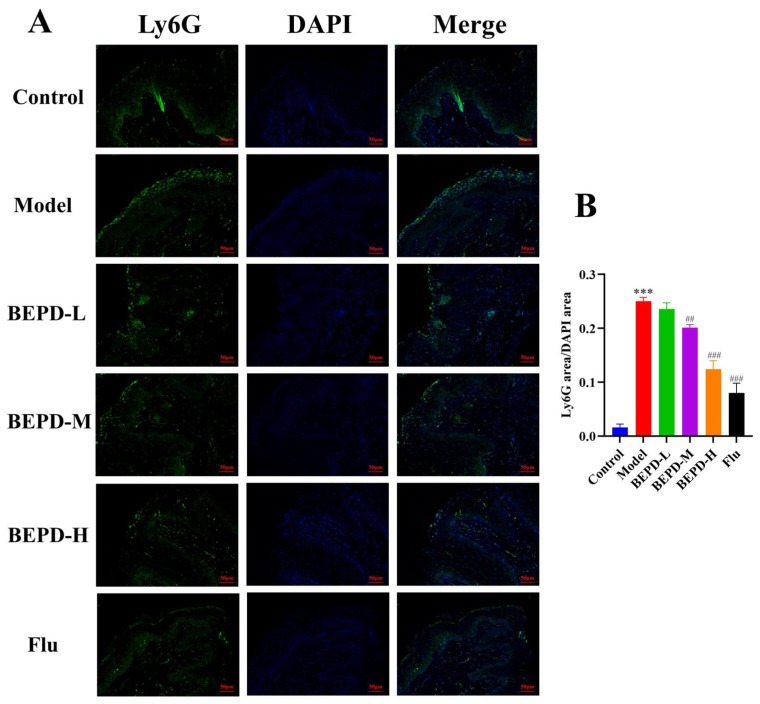
BEPD reduced neutrophil infiltration and aggregation (IF, 200×). (**A**) Images represent IF staining of the vagina. (**B**) The semi-quantification of Ly6G expression was performed using Image J (version ij153-win-java8). Values are presented as mean ± SD. n = 5. *** *p* < 0.001, vs. control group; ## *p* < 0.01, vs. model group; ### *p* < 0.001, vs. model group. BEPD, n-butanol extract of Pulsatilla decoction; BEPD-L, low-dose BEPD group (20 mL/kg); BEPD-M, medium-dose BEPD group (40 mL/kg); BEPD-H, high-dose BEPD group (80 mL/kg); Flu, fluconazole group; IF: immunofluorescence.

**Figure 11 pharmaceuticals-17-00594-f011:**
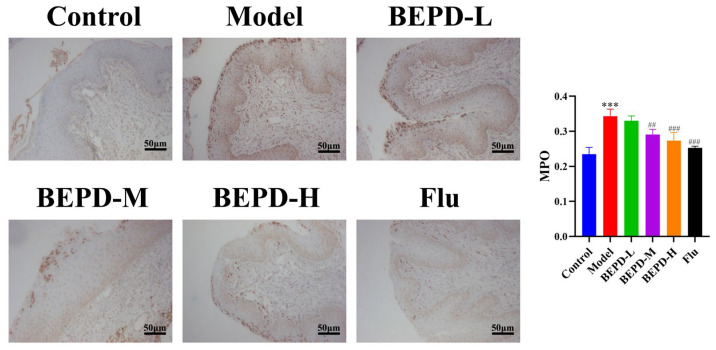
BEPD inhibited MPO expression in vaginal tissues of mice with VVC (IHC, 200×). Images represent IHC staining of the vagina. The semi-quantification of Ly6G expression was performed using Image J. Values are presented as mean ± SD. n = 3. *** *p* < 0.01, vs. control group; ## *p* < 0.01, vs. model group; ### *p* < 0.001, vs. model group. BEPD, n-butanol extract of Pulsatilla decoction; BEPD-L, low-dose BEPD group (20 mL/kg); BEPD-M, medium-dose BEPD group (40 mL/kg); BEPD-H, high-dose BEPD group (80 mL/kg); Flu, fluconazole group; VVC, vulvovaginal candidiasis; IHC: immunohistochemistry; MPO, myeloperoxidase.

**Figure 12 pharmaceuticals-17-00594-f012:**
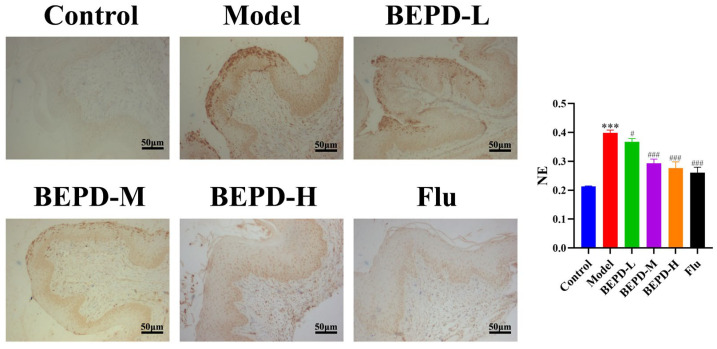
BEPD inhibited neutrophil elastase expression in vaginal tissues of mice with VVC (IHC, 200×). Images represent IHC staining of the vagina. The semi-quantification of Ly6G expression was performed using Image J. Values are presented as mean ± SD. n = 3. *** *p* < 0.01, vs. control group; # *p* < 0.05, vs. model group; ### *p* < 0.001, vs. model group. BEPD, n-butanol extract of Pulsatilla decoction; BEPD-L, low-dose BEPD group (20 mL/kg); BEPD-M, medium-dose BEPD group (40 mL/kg); BEPD-H, high-dose BEPD group (80 mL/kg); Flu, fluconazole group; VVC, vulvovaginal candidiasis; IHC: immunohistochemistry; NE, neutrophil elastase.

**Figure 13 pharmaceuticals-17-00594-f013:**
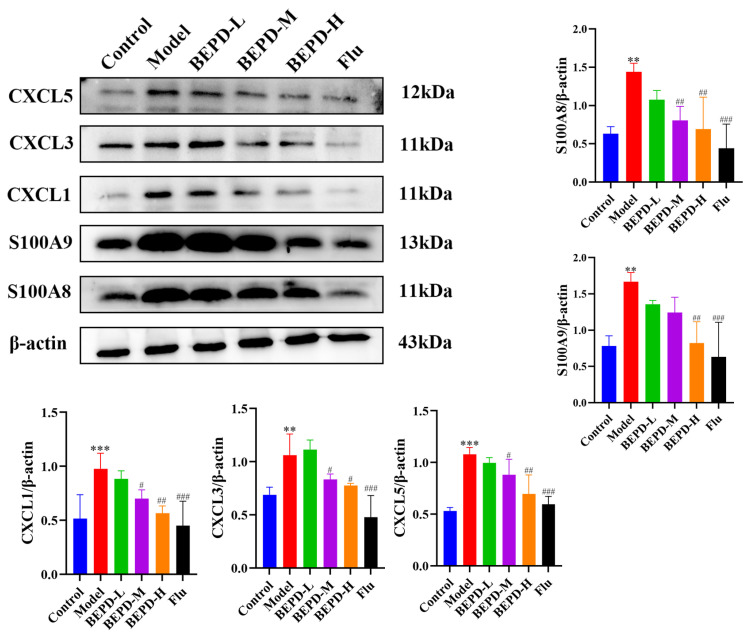
BEPD downregulated the expression of chemokine-associated proteins (Western blots). Western blots were utilized to detect the expression of CXCL1, CXCL3, S100A8, and S100A9 proteins. Values are presented as mean ± SD. *n* = 3. ** *p* < 0.01, vs. control group; *** *p* < 0.01, vs. control group; # *p* < 0.05, vs. model group; ## *p* < 0.01, vs. model group; ### *p* < 0.001, vs. model group. BEPD, n-butanol extract of Pulsatilla decoction; BEPD-L, low-dose BEPD group (20 mL/kg); BEPD-M, medium-dose BEPD group (40 mL/kg); BEPD-H, high-dose BEPD group (80 mL/kg); Flu, fluconazole group.

**Figure 14 pharmaceuticals-17-00594-f014:**
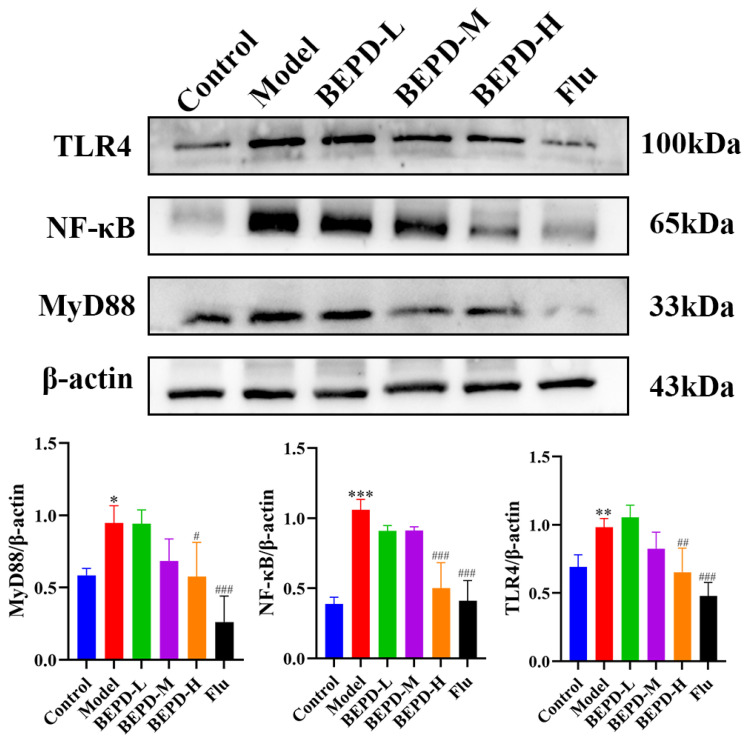
VVC is treated by BEPD, which inhibits the activation of the TLR4 signaling pathway (Western blots). Western blots were utilized to detect the expression of TLR4, MyD88, and NF-κB proteins. Values are presented as mean ± SD. *n* = 3. * *p* < 0.05, vs. control group; ** *p* < 0.01, vs. control group; *** *p* < 0.001, vs. control group; # *p* < 0.05, vs. model group; ## *p* < 0.01, vs. model group; ### *p* < 0.001, vs. model group. BEPD, n-butanol extract of Pulsatilla decoction; BEPD-L, low-dose BEPD group (20 mL/kg); BEPD-M, medium-dose BEPD group (40 mL/kg); BEPD-H, high-dose BEPD group (80 mL/kg); VVC, vulvovaginal candidiasis; Flu, fluconazole group; MyD88, myeloid differentiation primary response gene 88; TLR, Toll-like receptor.

**Figure 15 pharmaceuticals-17-00594-f015:**
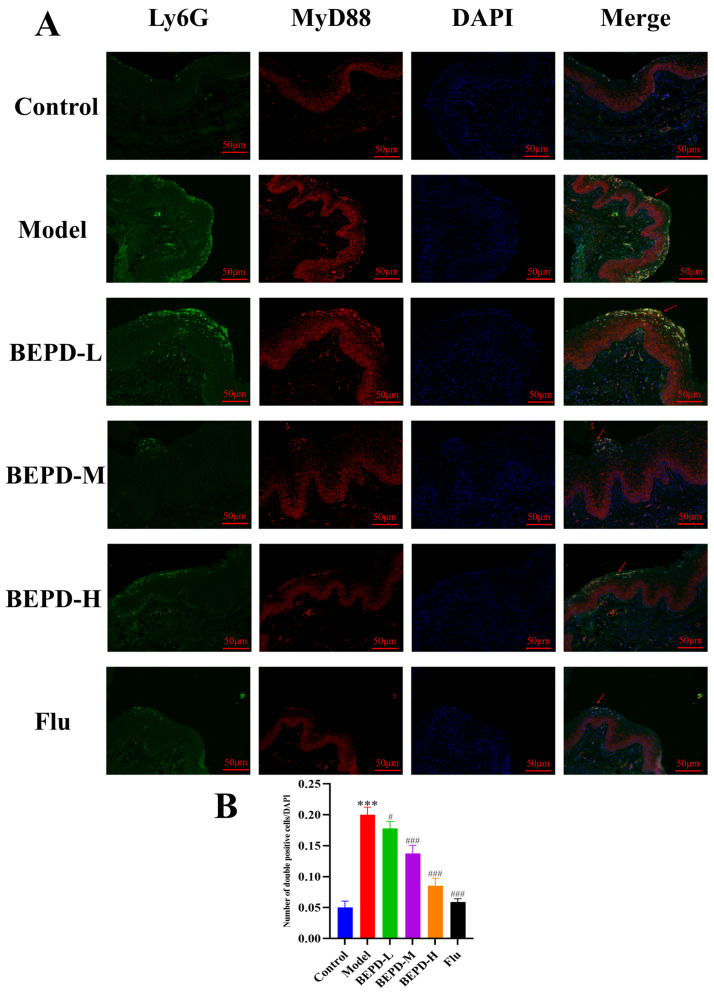
BEPD reduced the co-calibration rate of Ly6G with MyD88 (IF, 200×). (**A**) Images represent IF staining of the vagina. (**B**) The co-expression of Ly6G and MyD88 was semi-quantitatively analyzed by Image J. The red arrows mark the co-expression sites of Ly6G and MyD88, and the larger the yellow area, the more the two are co-expressed. Values are presented as mean ± SD. n = 3. *** *p* < 0.001, vs. control group; # *p* < 0.05, vs. model group; ### *p* < 0.001, vs. model group. BEPD, n-butanol extract of Pulsatilla decoction; BEPD-L, low-dose BEPD group (20 mL/kg); BEPD-M, medium-dose BEPD group (40 mL/kg); BEPD-H, high-dose BEPD group (80 mL/kg); Flu, fluconazole group; MyD88, myeloid differentiation primary response gene 88; IF: immunofluorescence.

**Figure 16 pharmaceuticals-17-00594-f016:**
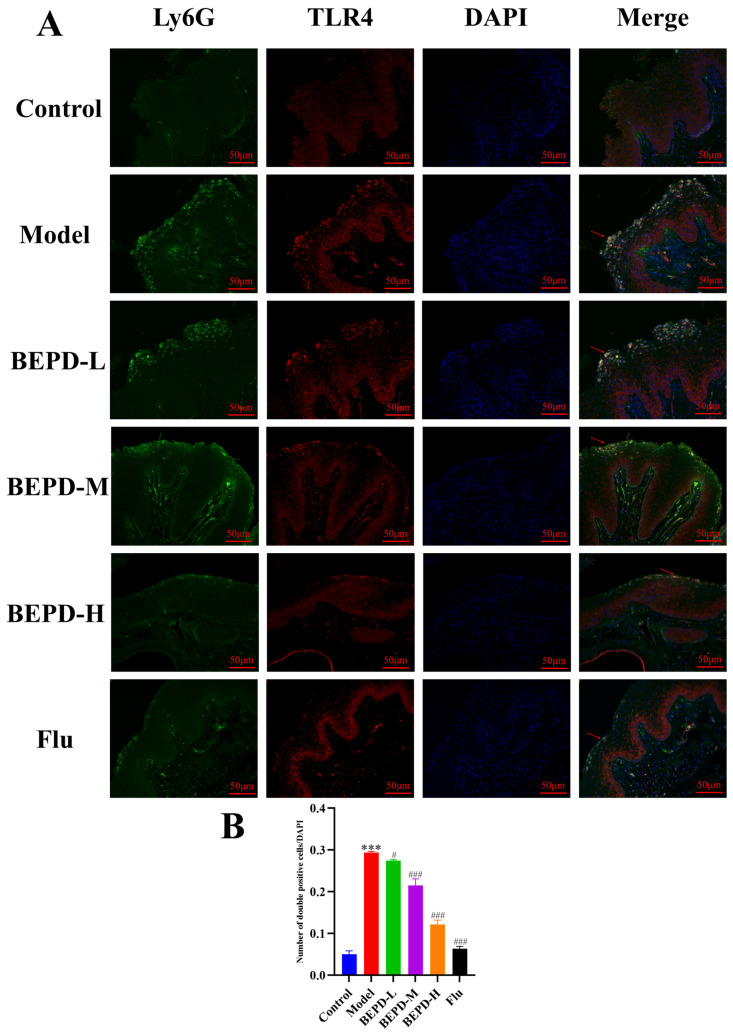
BEPD reduced the co-calibration rate of Ly6G with TLR4 (IF, 200×). (**A**) Images represent IF staining of the vagina. (**B**) The co-expression of Ly6G and MyD88 was semi-quantitatively analyzed by Image J. The red arrows mark the co-expression sites of Ly6G and TLR4, and the larger the yellow area, the more the two are co-expressed. Values are presented as mean ± SD. n = 3. *** *p* < 0.001, vs. control group; # *p* < 0.05, vs. model group; ### *p* < 0.001, vs. model group. BEPD, n-butanol extract of Pulsatilla decoction; BEPD-L, low-dose BEPD group (20 mL/kg); BEPD-M, medium-dose BEPD group (40 mL/kg); BEPD-H, high-dose BEPD group (80 mL/kg); Flu, fluconazole group; TLR, Toll-like receptor; IF: immunofluorescence.

**Figure 17 pharmaceuticals-17-00594-f017:**
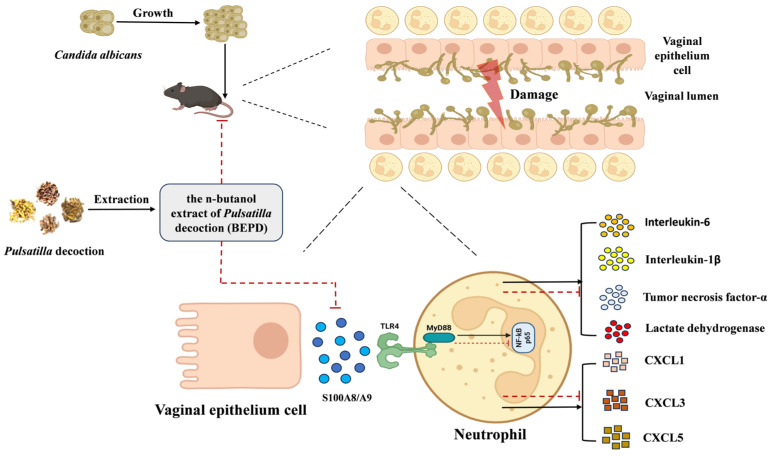
Schematic illustration of this study. BEPD regulates neutrophil recruitment and chemotaxis by inhibiting the release of S100A8/A9 and downregulating the TLR4/MyD88/NF-κB signaling pathway. Consequently, the release of IL-1β, IL-6, TNF-α, and LDH, as well as chemokines (CXCL1, CXCL3 and CXCL5), is reduced, which ultimately protects the vaginal mucosa of VVC mice.

**Table 1 pharmaceuticals-17-00594-t001:** The retention time, peak areas, and contents of anemoside B4, phellodendrine, esculin, esculetin, epiberberine, berberine, and jatrorrhizine in BEPD. BEPD: n-butanol extract of Pulsatilla decoction.

Name	Retention Time (min)	Peak Area (mAU×min)	Content (ug/g)
Anemoside B4	12.748	17.402	41,196.0197
Phellodendrine	12.803	7.650	7564.5979
Esculin	6.374	269.097	67,965.3986
Esculetin	11.821	0.364	83.8578
Epiberberine	12.047	5.78	1852.2891
Berberine	18.013	94.903	66,113.7401
Jatrorrhizine	18.496	92.993	15,401.7182

**Table 2 pharmaceuticals-17-00594-t002:** Primer sequences used for RT-qPCR.

Gene	Primer Sequences
mouse- CXCL1	Forward (5′-3′): ACCCGCTCGCTTCTCTGTG
	Reverse (5′-3′): GCTCTGGATGTTCTTGAGGTGAATC
mouse- CXCL3	Forward (5′-3′): ACTGGTCCTGCTGCTGCTG
	Reverse (5′-3′): TCACCGTCAAGCTCTGGATGG
mouse- CXCL5	Forward (5′-3′): CGGTTCCATCTCGCCATTCATG
	Reverse (5′-3′): GGAGTTACGGTTAAGCAAACACAAC
mouse- S100A8	Forward (5′-3′): TGCCCTCTACAAGAATGACTTCAAG
	Reverse (5′-3′): TTTATCACCATCGCAAGGAACTCC
mouse- S100A9	Forward (5′-3′): CGCAGCATAACCACCATCATCG
	Reverse (5′-3′): AGGGCTTCATTTCTCTTCTCTTTCTTC
mouse- GAPDH	Forward (5′-3′): GAAGCAGGGATTAAAGTG
	Reverse (5′-3′): TTCTTCTCGAAACCCTGATA

## Data Availability

The authors confirm that the data supporting the findings of this study are available within the article.

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
