# Peer review of "Transcriptomics Reveals Effect of Pulsatilla Decoction Butanol Extract in Alleviating Vulvovaginal Candidiasis by Inhibiting Neutrophil Chemotaxis and Activation via TLR4 Signaling"

_pharmaceuticals, 2024, doi:10.3390/ph17050594_

Round 1

Reviewer 1 Report

Comments and Suggestions for Authors

The manuscript by Wu et al. presents a study that offers valuable insights into the therapeutic potential of n-butanol extract of Pulsatilla decoction (BEPD) for VCC treatment. The study demonstrated the effectiveness of BEPD in reducing VVC symptoms and protecting vaginal tissues. It achieved this by suppressing neutrophil chemotaxis and activation, as well as regulating the TLR4/MyD88/NF-κB pathway.

The manuscript has presented valuable information.  The experiments are meticulously designed.  I have a few suggestions for polishing the manuscript.

Major Comments:

  1. The manuscript should be checked by a native English speaker. It needs some minor editing.
  2. There are certain figure legends that were explained in only one sentence. The figure legends should be discussed elaborately.
  3. The authors should state whether they observed any change in the body weight of mice in Figure 2.
  4. The authors performed gram staining in order to state the differences between the quantity of hyphal formation in control, model, and treated mice with VVS (Fig. 3). Although the results are prominent, the authors should perform SEM to show hyphal distribution. The authors should also quantify the amount of hyphal distribution.
  5. The authors should include proper scales in Figs. 5, 6, 10, 11, and 15.
  6. The authors should re-validate their RNA-seq data with proper qPCR analysis. They should quantify the relative steady-state levels of several chemokine and S100 genes.
  7. The authors should quantify the co-expression of Ly6G with TLR4 and MyD88 in Figure 15.

Minor Comments:

  1. In the introduction section, the authors should also provide some information about the effectiveness of Pulsatilla decoction against various human diseases. It will highlight the effectiveness of Pulsatilla decoction against a wide range of diseases.
  2. The authors may include a graphical abstract to depict the novelty of the study for a broader audience.
Comments on the Quality of English Language
  1. The manuscript should be checked by a native English speaker. It needs some minor editing.

Author Response

Dear Editor and Reviewers,

We are deeply grateful for the time and effort spent by the reviewers to improve our manuscript entitled “Transcriptomics reveals the effect of butanol extract of Pulsatilla decoction in alleviating vulvovaginal candidiasis by inhibiting neutrophil chemotaxis and activation via TLR4 signaling.” (Manuscript ID: 2980546). Their comments have been invaluable in helping us to revise and improve our paper, and they have also served as an important guide for our research. We have carefully considered the valuable comments provided by the reviewers and implemented the necessary changes. We hope to receive their approval. The revised portion is marked in red in the paper. The main corrections in the paper and the response to the reviewer’s comments are as flowing:

Reviewer1(s)' Comments to Author:

  • manuscript should be checked by a native English speaker. It needs some minor editing.

Response:

We are very grateful to the reviewer for their linguistic comment. Prior to submission, the English presentation of the thesis was first reviewed by each author included in the thesis, followed by approval from the supervisor. Thereafter, the thesis was handed over to a professional organisation for final editing. However, in response to your suggestion, we have conducted a further review of the language presentation of the thesis and hope to meet your expectations.

  • There are certain figure legends that were explained in only one sentence. The figure legends should be discussed elaborately.

Response:

We sincerely thank the reviewer for their comment.Upon review of the legend explanations in the paper, we have identified that the explanations of Figures 5, 6 and 15 were somewhat concise. To address this, we have revised the paper by expanding the contents of the legend explanations.

  • The authors should state whether they observed any change in the body weight of mice in Figure 2.

Response:

We are very grateful to the reviewer for their question. In many disease experiments, the weight change of mice is often used as a reference index. However, during the long-term study of VVC in our group, we found that the effect of VVC on the weight change of mice was minimal and almost negligible. Therefore, the weight of mice was not recorded. This can also be corroborated by previous publications on VVC (e.g. Hu K, Jiang X, Zhang J, et al. Effect of Pulsatilla decoction on vulvovaginal candidiasis in mice. Evidences for its mechanisms of action. Evidences for its mechanisms of action. Phytomedicine. Published online March 8, 2024), thus justifying the decision not to record the body weight changes in mice in this experiment.

  • The authors performed gram staining in order to state the differences between the quantity of hyphal formation in control, model, and treated mice with VVS (Fig. 3). Although the results are prominent, the authors should perform SEM to show hyphal distribution. The authors should also quantify the amount of hyphal distribution.

Response:

We thank the reviewer for their valuable suggestion on the experiments of this thesis. In this thesis, we applied Gram staining for the following purposes: firstly, to determine whether there is hyphal formation by staining Candida albicans in the lavage fluid. This is because the transformation of Candida albicans from the yeast state to the hypha state is one of the important characteristics of its pathogenicity. It also provides us with an indicator for determining the success of the model in mice at the stage of model. Secondly, at the stage of drug administration, it is possible to initially judge whether the drug has a therapeutic effect through the hypha, which provides the basis for a deeper exploration of drug therapy at a later stage. Therefore, it was not deemed necessary to use scanning electron microscopy to show the hyphal distribution and quantify the amount of hyphal distribution.

  • The authors should include proper scales in Figs. 5, 6, 10, 11, and 15.

Response:

We are grateful for the opinion of reviewer. In order to address the issue of scale (μm) in microscopic images as previously mentioned, we have added a scale bar to the bottom right corner of the image.

  • The authors should re-validate their RNA-seq data with proper qPCR analysis. They should quantify the relative steady-state levels of several chemokine and S100 genes.

Response:

We would like to express our gratitude to the reviewer for their invaluable feedback on the experimental refinement of this paper. We have validated the data in RNA-seq using qPCR and quantified the results. Our findings indicate that the expression of CXCL1, CXCL3, CXCL5, S100a8, and S100a9 mRNA was elevated in the model group, while it was significantly reduced after BEPD treatment.

  • The authors should quantify the co-expression of Ly6G with TLR4 and MyD88 in Figure 15.

Response:

We thank the reviewer for their comment. We have quantified the co-expression of Ly6G with TLR4 and MyD88 and added it to the paper.

  • In the introduction section, the authors should also provide some information about the effectiveness of Pulsatilladecoction against various human diseases. It will highlight the effectiveness of Pulsatilladecoction against a wide range of diseases.

Response:

We are very grateful to the reviewer for their suggestion. We have added information about the efficacy of Pulsatilla decoction for various human diseases in the introductory section of the paper, specifically in lines 83 to 87 of the paper.

  • The authors may include a graphical abstract to depict the novelty of the study for a broader audience.

Response:

We are very grateful to the reviewer for their comment.The graphical abstract of the paper is both intuitive and concise, effectively summarising the paper's exposition. We have realised the importance of this and have added it to the paper, which we hope will be approved by you.

Reviewer 2 Report

Comments and Suggestions for Authors

Dear Authors,

The submitted Manuscript is about the study of butanol extract of Pulsatill decoction in the control of vulvovaginal candidiasis. However, it requires necessary revisions before it can be accepted for publication. These mainly concern the Abstract and the Introduction, which are written in a vague manner (Abstract) and too poorly (Introduction).

Affiliation - please remove the links and use the style required by MDPI

Abstract - it is necessary to write it from scratch. In its current form, it does not introduce the reader sufficiently to the topic covered. The first two sentences can be combined, they are about the same thing. For the sake of order, please say separately: purpose, methods, results and conclusions. For now, after reading, the reader has runaway thoughts and does not know what to expect in the Manuscript. The abstract should be written in a clear, lucid and interesting way. It should encourage the reader to read the Article. I found it completely uninteresting.

Introduction: It should introduce the topic under study and indicate why it is so important that it is worth studying. What benefits this research will bring us. Please emphasize this in more detail. To the 1st paragraph, the authors should add what are the consequences of infection with C. albicans etiology. It is worth adding that it is an opportunistic pathogen, i.e. that its infection is associated with certain predispositions - list which ones. In the last paragraph, please expand on the information regarding the preparation used by the authors. Please briefly discuss the % composition of each ingredient mentioned and add more information about it, e.g., from what it is obtained, what active compounds predominate in it, what are its most important activities. Such information is important for those researchers who will want to study individual ingredients. 

Table 1 - please confirm whether the units used are correct

Figures - under each please explain the abbreviations used

Figures 7,8 - are illegible, please enlarge

Figure 15 - please describe it in more detail directly below it. What is located on 15A and what is on 15B. Mention what the specific fluorescent dye was used for. 

Line 317 - "Candida albicans" - there is no need for the full name, as this is not its first use in the Manuscript.

Best regards

Author Response

Dear Editor and Reviewers,

We are deeply grateful for the time and effort spent by the reviewers to improve our manuscript entitled “Transcriptomics reveals the effect of butanol extract of Pulsatilla decoction in alleviating vulvovaginal candidiasis by inhibiting neutrophil chemotaxis and activation via TLR4 signaling.” (Manuscript ID: 2980546). Their comments have been invaluable in helping us to revise and improve our paper, and they have also served as an important guide for our research. We have carefully considered the valuable comments provided by the reviewers and implemented the necessary changes. We hope to receive their approval. The revised portion is marked in red in the paper. The main corrections in the paper and the response to the reviewer’s comments are as flowing:

Reviewer2(s)' Comments to Author:

Comment 1:Affiliation - please remove the links and use the style required by MDPI.

Response:

Many thanks to the reviewer for their comment. We have corrected the errors in the paper according to the format.

Comment 2:Abstract - it is necessary to write it from scratch. In its current form, it does not introduce the reader sufficiently to the topic covered. The first two sentences can be combined, they are about the same thing. For the sake of order, please say separately: purpose, methods, results and conclusions. For now, after reading, the reader has runaway thoughts and does not know what to expect in the Manuscript. The abstract should be written in a clear, lucid and interesting way. It should encourage the reader to read the Article. I found it completely uninteresting.

Response

We would like to express our sincerest gratitude to the reviewer for the valuable suggestion regarding the language of the paper. We have undertaken a comprehensive revision of the abstract, striving to enhance its clarity and conciseness. By adhering to the order of research objectives, methods, results, and conclusions, we have endeavored to make the abstract more coherent and accessible. We hope that this revision meets with your satisfaction. Finally, we would like to reiterate our profound gratitude to you for your valuable suggestion.

Comment 3:Introduction: It should introduce the topic under study and indicate why it is so important that it is worth studying. What benefits this research will bring us. Please emphasize this in more detail. To the 1st paragraph, the authors should add what are the consequences of infection with C. albicans etiology. It is worth adding that it is an opportunistic pathogen, i.e. that its infection is associated with certain predispositions - list which ones. In the last paragraph, please expand on the information regarding the preparation used by the authors. Please briefly discuss the % composition of each ingredient mentioned and add more information about it, e.g., from what it is obtained, what active compounds predominate in it, what are its most important activities. Such information is important for those researchers who will want to study individual ingredients. 

Response

We would like to express our gratitude for your valuable comment. We have addressed the issues you raised in the introduction by rewriting the first and last paragraphs. We conducted a review of the relevant literature and concluded that Candida albicans infection is closely related to the long-term use of broad-spectrum antibiotics, glucocorticosteroids or immunosuppressants. We have also included a description of the clinical symptoms of VVC in the first paragraph of the introduction. In the final paragraph, we elucidated the role of the four herbs and the related chemical constituents in Pulsatilla decoction. For the principal active ingredients in the n-butanol extract of Pulsatilla decoction, The assay was performed by HPLC, and the individual active ingredients were summarised in the results. We hope that the revised introduction will be approved by you.

Comment 4:Table 1 - please confirm whether the units used are correct.

Response:

We are very grateful to the reviewer for their comment. The units used in Table 1 were provided from the company we tested. We have compared them one by one and confirmed that there is no problem. We can provide the test report if there is a subsequent need.

Comment 5:Figures - under each please explain the abbreviations used.

Response:

We are very grateful to the reviewer for their linguistic suggestion. We have reinterpreted each abbreviation of the figures provided in the paper. Thank you again for your valuable suggestions.

Comment 6:Figures 7,8 - are illegible, please enlarge

Response:

We are very grateful to the reviewer for their suggestion on the issue of image layout. For the problem that the pictures in Figures 7 and 8 are too small, we have adopted the method of splitting the whole picture and combining the two small pictures into one large one, and rearranged the layout, and we hope that the size of the pictures can meet your requirements.

Comment 7:Figure 15 - please describe it in more detail directly below it. What is located on 15A and what is on 15B. Mention what the specific fluorescent dye was used for. 

Response:

We appreciate the reviewer's question. For Figure 15, firstly, we have quantified and split it, changing the original Figure 15A to Figure 15 and the original Figure 15B to Figure 16, secondly, the colocalization of Ly6G with TLR4 and MyD88 has been marked on Figure 15 and Figure 16 with red arrows, after the colocalization of the two proteins, the corresponding fluorescence overlaps to form a yellow spotting , and the larger the yellow area, the more the two proteins were co-expressed.Sections of vaginal tissue were immunostained with DAPI (blue) , FITC-anti-Ly6G (green) , AF594-anti-TLR4 (red) and AF594-anti-MyD88 (red) observed by the Leica fluorescence microscope on fluorescent dyes.

Comment 8:Line 317 - "Candida albicans" - there is no need for the full name, as this is not its first use in the Manuscript.

Response:

We are truly grateful for the linguistic help of reviewer. We have corrected the errors in the revised manuscript. Thanks again for your valuable suggestions.

Round 2

Reviewer 1 Report

Comments and Suggestions for Authors

The manuscript by Wu et al. presents a study that offers valuable insights into the therapeutic potential of n-butanol extract of Pulsatilla decoction (BEPD) for VCC treatment. The study demonstrated the effectiveness of BEPD in reducing VVC symptoms and protecting vaginal tissues. It achieved this by suppressing neutrophil chemotaxis and activation, as well as regulating the TLR4/MyD88/NF-κB pathway.

The authors have addressed all the previous comments. Thus, the manuscript can be accepted in its present form.

Reviewer 2 Report

Comments and Suggestions for Authors

Dear Authors,

In my opinion, the Manuscript is suitable for publication

Best regards